# Design, Synthesis and Characterization of a New Series of Fluorescent Metabotropic Glutamate Receptor Type 5 Negative Allosteric Modulators

**DOI:** 10.3390/molecules25071532

**Published:** 2020-03-27

**Authors:** Víctor Fernández-Dueñas, Mingcheng Qian, Josep Argerich, Carolina Amaral, Martijn D.P. Risseeuw, Serge Van Calenbergh, Francisco Ciruela

**Affiliations:** 1Unitat de Farmacologia, Departament de Patologia i Terapèutica Experimental, Facultat de Medicina i Ciències de la Salut, IDIBELL, Universitat de Barcelona, 08907 L’Hospitalet de Llobregat, Spain; jargerich95@gmail.com (J.A.); ccarolinaamaral@gmail.com (C.A.); 2Institut de Neurociències, Universitat de Barcelona, 08035 Barcelona, Spain; 3Laboratory for Medicinal Chemistry (FFW), Ghent University, Ottergemsesteenweg 460, B-9000 Ghent, Belgium; mqian2019@cczu.edu.cn (M.Q.);; 4Laboratory of Toxicology, Ghent University, Ottergemsesteenweg 460, B-9000 Ghent, Belgium

**Keywords:** mGlu_5_R, fluorescent ligands, allosterism, nanoBRET

## Abstract

In recent years, new drug discovery approaches based on novel pharmacological concepts have emerged. Allosteric modulators, for example, target receptors at sites other than the orthosteric binding sites and can modulate agonist-mediated activation. Interestingly, allosteric regulation may allow a fine-tuned regulation of unbalanced neurotransmitter’ systems, thus providing safe and effective treatments for a number of central nervous system diseases. The metabotropic glutamate type 5 receptor (mGlu_5_R) has been shown to possess a druggable allosteric binding domain. Accordingly, novel allosteric ligands are being explored in order to finely regulate glutamate neurotransmission, especially in the brain. However, before testing the activity of these new ligands in the clinic or even in animal disease models, it is common to characterize their ability to bind mGlu_5_Rs in vitro. Here, we have developed a new series of fluorescent ligands that, when used in a new NanoBRET-based binding assay, will facilitate screening for novel mGlu_5_R allosteric modulators.

## 1. Introduction

G protein-coupled receptors (GPCRs) are a superfamily of transmembrane receptors that detect and transmit a large array of extracellular signals (i.e., sensory stimuli, hormones, neurotransmitters), which allow regulating many different physiological functions (i.e., vision, blood pressure, central nervous system activity) [1]. Accordingly, these kinds of receptors represent the main target (~35%) of clinically used drugs [2]. Most GPCR-targeting drugs consist of ligands, either agonists or antagonists, which bind to the endogenous ligand (orthosteric) binding site. However, in recent years, an emerging alternative approach to develop novel drugs consists of searching for allosteric ligands, which: (i) bind to sites topographically distinct from the orthosteric one, and (ii) typically, do not possess intrinsic activity, but only modulate the endogenous ligand-mediated function [3]. There exist two kinds of allosteric modulators, positive and negative allosteric modulators (PAM and NAM, respectively) [3]. Both PAM and NAM may offer a number of advantages over typical orthosteric ligands. For instance, these drugs have the potential for higher receptor subtype selectivity, since the allosteric binding pockets present higher sequence diversity compared to their orthosteric counterparts, which are more conserved and are thus likely to trigger off-target effects [4,5]. However, the main strength of allosteric modulators consists of their inability of activating receptors by their own. The activity of these kinds of drugs depends on endogenous ligands-mediated signaling, thus they may finely modulate receptors instead of fully activating/blocking them [5]. Interestingly, the fine-tuning regulation of receptors may be especially relevant in pathologies in which neurotransmission systems are unbalanced. This is the case, for instance, for benzodiazepines, which provide a safe (with a high therapeutic index) and effective treatment for anxiety, acting as PAMs for the GABA_A_ receptor [6]. 

Based on the previous notions, in recent years, a number of receptor subtype-selective allosteric ligands have been developed for different neurotransmitters’ receptors [7,8,9,10], such as the metabotropic glutamate receptors (mGluR), which are involved in numerous central nervous system pathologies (i.e., schizophrenia, pain, depression, Parkinson’s disease) [11,12,13]. Glutamate is the major excitatory neurotransmitter within the central nervous system and exerts its effects by interacting with two main types of receptors (ionotropic and metabotropic) [14,15]. There are eight subtypes of metabotropic receptors, which are divided into three groups (I, II, III) based on their homology, G-protein coupling and pharmacological profile [14,15]. Group I, comprising mGlu_1_R and mGlu_5_R, are coupled to G_q_/G_11_ proteins, and upon stimulation lead to phospholipase C activation, IP_3_ accumulation, and Ca^2+^ release from intracellular stores [15]. These receptors are characterized by a large N-terminal domain, termed as the venus-flytrap domain, which contains the endogenous ligand-binding site; while the allosteric binding site(s) is located at the transmembrane domain [5]. Interestingly, to screen for novel mGlu_1_R and mGlu_5_R PAMs or NAMs, radioligands and positron emission tomography (PET) ligands have been developed [5]. These kinds of ligands allow performing cell-based assays, in which apart from assessing orthosteric agonist modulation, it is possible to perform binding competition assays to assess affinity for the allosteric binding site. However, both PET and radioligand binding are expensive, time-consuming and/or hazardous techniques, thus implementing fluorescence-based tools could be desirable. The main drawback of using fluorescent mGlu_5_R allosteric ligands is related to their high lipophilic nature, which may cause adsorption to cell surfaces [15]. To circumvent this drawback, resonance energy transfer (RET) techniques can be implemented [16,17]. 

Here, we have developed a new approach to screen for mGlu_5_R allosteric ligands, based on the engagement of a BRET process between mGlu_5b_R labelled with a nanolucifecrase (i.e., NanoLuc) and selective fluorescent allosteric ligands. Accordingly, we have designed and synthetized a new series of fluorescent mGlu_5_R NAMs, which may be useful to develop a high-throughput screening for novel mGlu_5_R allosteric ligands.

## 2. Results

### 2.1. Design and Synthesis of the Fluorescent Ligands

Boron dipyrromethene (BODIPY) is a fluorescent dye with diverse properties, such as photochemical stability, high molar absorptivity, high fluorescence quantum yield, and the fact that its fluorescence property can be altered by varying the substitution pattern on the core and the flanking pyrroles. Alkyne functionalized BODIPY dyes can be used to fluorescently label an azido functionalized ligand in a chemo selective manner employing the copper-catalyzed azide-alkyne cycloaddition (CuAAC) reaction [18]. Here, we exploited alkyne functionalized BODIPY (573/607) to construct a small set of fluorescent mGlu_5_R ligands based on MTEP (Figure 1). 

The mGlu_5_R carboxylate derivatives were condensed with different spacers containing both an amine and azide terminus in the presence of the coupling agent EDC and triethylamine to yield six intermediate azides as described previously [19]. The synthesis of the desired fluorescent ligands was accomplished by reacting each of these azides with an alkyne-modified BODIPY red dye through CuAAC. This afforded a concise series of six mGlu_5_R fluorescent ligands **4a**–**f** (Figure 2).

The excitation and emission spectra of the fluorescent mGlu_5_R ligands was determined. All ligands exhibited similar excitation and emission wavelengths with comparable spectral separation (~40 nm) between them (Figure 3a). Next, we assessed the fluorescence intensity by exciting ligands at 573 nm (the theoretical peak of excitation of BODIPY) and at 490 nm (the emission wavelength of NanoLuc). We used the same concentration of each ligand (10 µM) and read fluorescence intensity at 607 nm (the theoretical peak of emission of BODIPY) and 610 nm (the wavelength to be read after the BRET process between NanoLuc and the fluorescent ligands), respectively. As shown in Figure 3b, ligand **4a** exhibited a significantly higher fluorescence emission intensity compared to the other ligands. Notably, the high fluorescence emission intensity observed at 610 nm upon 490 nm excitation supported the possibility of using **4a** in the subsequent nanoBRET binding assay. In addition, we also selected **4e** to be used in further experiments, based on the fact that it exhibited the second highest fluorescence emission intensity at both wavelengths (Figure 3b).

### 2.2. Characterization of the NanoBRET Binding Assay

To set up the nanoBRET binding assay, we first validated the cloning of the mGlu_5b_R^NL^ construct. Using immunoblot analysis, mGlu_5b_R^NL^ expression was ascertained by the presence of a protein band of molecular weight ~150 kDa, corresponding to the sum of mGlu_5b_R (~130 kDa) and NanoLuc (~19 kDa) proteins (Figure 4a). Moreover, we ascertained the luminescence of cells expressing the mGlu_5b_R^NL^, by incubating these with the enzyme’s substrate (coelenterazine 400a, 1 µM). While in mock-transfected cells no signal was observed, a robust luminescent signal was obtained in mGlu_5b_R^NL^ upon incubation for 5 min with coelenterazine 400a (Figure 4b). Next, we evaluated whether the fluorescent ligands could bind and activate mGlu_5b_R, a condition sine qua non for using them in a nanoBRET binding assay. First, we imaged mGlu_5b_R^SNAP^-expressing cells (Figure 4c) superfused with the fluorescent ligand **4a**. Interestingly, **4a** nicely decorated the cell surface, but was also present inside the cell after 5 min of incubation (Figure 4d). The fact that the ligand penetrated into the cell, either by diffusion or endocytosis, supported the need for developing the nanoBRET binding assay instead of determining fluorescence intensity. 

Next, we tested whether the fluorescent ligands maintained their ability to allosterically modulate agonist-mediated mGlu_5_R signalling. Accordingly, cells were transfected with the mGlu_5b_R^SNAP^ construct and a NFAT-luciferase reporter plasmid (pGL4-NFAT-RE/luc2p) to indirectly determine intracellular Ca^2+^ accumulation (Figure 5a). In these cells, activation of mGlu_5b_R via the application of the agonist quisqualic acid (100 μM) increased intracellular Ca^2+^, which enhanced NFAT-sensitive expression of the luciferase reporter (Figure 5b). Conversely, intracellular Ca^2+^ accumulation was blocked when cells were co-incubated with quisqualic acid (100 μM) and either the prototypic mGlu_5_R NAM MTEP or ligands **4a** and **4e** (Figure 5b). It can thus be concluded that the fluorescent ligands tested retained the ability to bind and allosterically modulate mGlu_5_R activity.

### 2.3. NanoBRET Binding Assay

Finally, we performed the nanoBRET binding assay (Figure 6a) by transfecting cells with the mGlu_5b_R^NL^ construct and challenging increasing concentrations of the fluorescent ligands, in the presence/absence of the non-labeled MTEP. Interestingly, a binding saturation hyperbola was obtained for both **4a** (K_D_ = 0.84 ± 0.74 µM and B_max_ = 37.3 ± 11.4%) and **4e** (K_D_ = 1.4 ± 1.2 µM and B_max_ = 54.3 ± 18.5%), while in the presence of a saturating concentration of MTEP (10 µM) the binding was displaced (Figure 6b). Hence, our results show that, despite the high lipophilic nature of the fluorescent ligands, it is possible to engage a BRET process, thus supporting that this nanoBRET binding assay is a robust and reliable way to assess mGlu_5_R allosteric ligand binding.

## 3. Discussion

In recent years, different mGlu_5_R allosteric drugs have shown efficacy in preclinical animal models of disease (i.e., anxiety, depression, drug abuse) [13]. Indeed, a number of clinical trials have been engaged, but, to our knowledge, most studies have failed to translate preclinical findings into clinics (for review see [21,22]). However, it is important to note that in most cases (i.e., basimglurant in depression), the inconclusive data obtained in clinical trials may be explained by some of the intrinsic issues of these kinds of trials, such as the placebo effect or limitations related to the treatment duration or to the selection of a single dose [22]. Accordingly, in the next years, designing better clinical trials is a major challenge to develop improved mGlu_5_R allosteric drugs. As there remains a high interest in developing these kinds of drugs, novel tools to rapidly characterize allosteric ligands are clearly awaited. Here, we have designed, synthesized, and characterized a novel series of fluorescent mGlu_5_R NAMs, which can be used in a nanoBRET assay to robustly and reliably assess mGlu_5_R allosteric ligand binding. This assay based on RET overcomes some of the drawbacks of other typical techniques, such as PET or radioligand binding. For instance, PET ligands can be extremely useful to visualize disruptions in glutamate transmission or to select the dose to be used in clinical trials, but they are expensive to perform ligand binding competition assays. Similarly, radioligand binding assays may provide robust data concerning pharmacological constants, but they are also expensive, harmful, time-consuming and difficult to miniaturize [16,23]. Regarding RET-based assays, a number of approaches have been implemented to study orthosteric ligand-receptor binding. For instance, a FRET-based assay (i.e., Tag-lite binding assay) was successfully and reproducibly applied to test novel ligands for different GPCRs, demonstrating its suitability for high-throughput screening [23]. Similarly, a new BRET-based assay was recently developed using GPCRs tagged with a NanoLuc protein to engage a RET process with fluorescent ligands [17,24]. Interestingly, this assay was truly homogenous (without any washing step), since it did not require conjugating a fluorophore to the GPCR (as in the Tag-lite assay), and was extremely sensitive, due to the brightness of the NanoLuc luciferase [17,24]. In the present work, we took advantage of this observation to investigate allosteric ligand binding. Of note, both the non-washing conditions and high sensitivity of the nanoBRET binding approach allowed to perform this kind of assay even with highly lipophilic fluorescent mGlu_5_R allosteric ligands. Despite the fact that these molecules were rapidly adsorbed and penetrated the plasma membrane, thus accumulating into the cell, it was possible to detect a specific signal dependent on mGlu_5_R ligand binding. In addition, it is noteworthy that the fluorescent ligands maintained the ability to interact and modulate agonist-mediated responses, thus it could be discarded that the BODIPY dye affects ligand binding. On the other hand, it is important to note that we used BODIPY derivatives with excitation/emission peaks at 573/607 nm. The reason for this is that NanoLuc exhibits a high emission upon coelenterazine incubation, and this signal overlaps and masks specific BRET signals when using dyes with lower excitation/emission wavelengths (data not shown). Altogether, our nanoBRET binding assay permitted to robustly assess mGlu_5_R allosteric ligand binding and may be viewed as a useful tool to develop novel mGlu_5_R allosteric ligands.

## 4. Materials and Methods

### 4.1. Chemistry

All reactions described were performed under an N_2_ atmosphere and at ambient temperature, unless stated otherwise. All reagents and solvents were purchased from Sigma-Aldrich (Diegem, Belgium), Fisher Scientific (Merelbeke, Belgium), TCI Europe (Zwijndrecht, Belgium) or Apollo Scientific (Bredbury, Stockport, UK), and used as received. NMR solvents were acquired from Eurisotop (Saint-Aubin, France). ESI-HRMS spectra were measured with a Waters LCT Premier XE Mass spectrometer calibrated using leu-enkephalin as an external standard. LC-MS analyses were carried out on a Waters AutoPurification System equipped with PDA and ESI-MS detection and using a Waters CORTECS C18 Column (4.6 × 100 mm, 2.7 μm) and a water/acetonitrile/formic acid linear gradient system at a flow rate of 1.44 mL min^−1^.

General procedure 1: Copper mediated azide-alkyne cycloaddition. To a solution of the azido-linker modifies MTEP analogues [19] (1.0 eq.) in dimethylformamide (0.1 M) was added the alkyne-modified BODIPY-red (1.5 eq.), sodium ascorbate (1.0 eq., 0.5 M), CuSO_4_ (0.2 eq., 0.05 M), triethylamine (3.0 eq.) and a catalytic amount of tris[(1-benzyl-1,2,3-triazol-4-yl)methyl]amine. The reaction mixture was stirred overnight at room temperature in the dark under an argon atmosphere. The solvent was evaporated under reduced pressure and the residue was partioned between water and CH_2_Cl_2_. The organic fraction was washed with brine and dried over Na_2_SO_4_. The crude compound was purified by silica gel chromatography (NH_4_OH/MeOH/CH_2_Cl_2_, 1:5:94 v/v/v) to give the final compounds as blue solids.

*N**-(17-(4-(4-(5,5-difluoro-3,7-bis(4-methoxyphenyl)-5**H**-4λ^4^,5λ^4^-dipyrrolo [1,2-c:2’,1’-f][1,3,2]diazaborinin-10-yl)butyl)-1**H**-1,2,3-triazol-1-yl)-3,6,9,12,15-pentaoxaheptadecyl)-2-((5-((2-methylthiazol-4-yl)ethynyl)pyridin-3-yl)oxy)acetamide* (**4a**)

Blue solid, 48%. LC-HRMS: *t*_R_ = 7.99 min (10–100% MeCN, 15 min run), purity 95.4%. HRMS (ESI) *m*/*z*: calculated for C_54_H_63_BF_2_N_8_O_9_S [M + 2H]^2+^ 524.2244; found 524.2223. Calculated for C_54_H_62_BFN_8_O_9_S [M − F + H]^2+^ 514.2219; found 514.2209.

*N**-(2-(2-(2-(2-(4-(4-(5,5-difluoro-3,7-bis(4-methoxyphenyl)-5**H**-4λ^4^,5λ^4^-dipyrrolo[1,2-c:2’,1’-f][1,3,2]diazaborinin-10-yl)butyl)-1**H**-1,2,3-triazol-1-yl)ethoxy)ethoxy)ethoxy)-ethyl)-2-(3-((2-methylthiazol-4-yl)ethynyl)phenoxy)acetamide* (**4b**)

Blue solid, 52%. LC-HRMS: *t*_R_ = 7.62 min (10–100% MeCN, 15 min run), purity 96.8%. HRMS (ESI) *m*/*z*: calculated for C_51_H_56_BF_2_N_7_O_7_S [M + 2H]^2+^ 479.7006; found 479.6974. Calculated for C_51_H_55_BF_2_N_7_O_7_S [M + H]^+^ 958.3939; found 958.3953.

*N**-(5-((12-(4-(4-(5,5-difluoro-3,7-bis(4-methoxyphenyl)-5**H**-4λ^4^,5λ^4^-dipyrrolo[1,2-c:2’,1’-f][1,3,2]diazaborinin-10-yl)butyl)-1**H**-1,2,3-triazol-1-yl)dodecyl)(methyl)amino)pentyl)-2-(3-((2-methylthiazol-4-yl)ethynyl)phenoxy)acetamide* (**4c**)

Blue solid, 46%. LC-HRMS: *t*_R_ = 9.53 min (10–100% MeCN, 15 min run), purity 85.4%. HRMS (ESI) *m*/*z*: calculated for C_61_H_77_BF_2_N_8_O_4_S [M + 2H]^2+^ 533.2919; found 533.2914. Calculated for C_61_H_76_BFN_8_O_4_S [M − F + H]^2+^ 523.2893; found 523.2885.

*N**-(5-((12-(4-(4-(5,5-difluoro-3,7-bis(4-methoxyphenyl)-5**H**-4λ^4^,5λ^4^-dipyrrolo[1,2-c:2’,1’-f][1,3,2]diazaborinin-10-yl)butyl)-1H-1,2,3-triazol-1-yl)dodecyl)(methyl)amino)pentyl)-2-((5-((2-methylthiazol-4-yl)ethynyl)pyridin-3-yl)oxy)acetamide* (**4d**)

Blue solid, 39%. LC-HRMS: *t*_R_ = 6.86 min (10–100% MeCN, 15 min run), purity 90.8%. HRMS (ESI) *m*/*z*: calculated for C_60_H_76_BF_2_N_9_O_4_S [M + 2H]^2+^ 533.7895; found 533.7896. Calculated for C_60_H_75_BFN_9_O_4_S [M − F + H]^2+^ 523.7870; found 523.7874.

*N**-(5-((8-(4-(4-(5,5-difluoro-3,7-bis(4-methoxyphenyl)-5**H**-4λ^4^,5λ^4^-dipyrrolo[1,2-c:2’,1’-f][1,3,2]diazaborinin-10-yl)butyl)-1**H**-1,2,3-triazol-1-yl)octyl)(methyl)amino)pentyl)-2-((5-((2-methylthiazol-4-yl)ethynyl)pyridin-3-yl)oxy)acetamide* (**4e**)

Blue solid, 37%. LC-HRMS: *t*_R_ = 5.93 min (10–100% MeCN, 15 min run), purity 82.5%. HRMS (ESI) *m*/*z*: calculated for C_56_H_68_BF_2_N_9_O_4_S [M + 2H]^2+^ 505.7582; found 505.7570. Calculated for C_56_H_67_BFN_9_O_4_S [M − F + H]^2+^ 495.7557; found 495.7557.

*N**-(6-(4-(4-(5,5-difluoro-3,7-bis(4-methoxyphenyl)-5**H**-4λ^4^,5λ^4^-dipyrrolo[1,2-c:2’,1’-f][1,3,2]diazaborinin-10-yl)butyl)-1**H**-1,2,3-triazol-1-yl)hexyl)-2-((5-((2-methylthiazol-4-yl)ethynyl)pyridin-3-yl)oxy)acetamide* (**4f**)

Blue solid, 45%. LC-HRMS: *t*_R_= 7.32 min (10–100% MeCN, 15 min run), purity 81.5 %. HRMS (ESI) *m*/*z*: calculated for C_48_H_51_BF_2_N_8_O_4_S [M + 2H]^2+^ 442.1902; found 442.1881. Calculated for C_48_H_50_BF_2_N_8_O_4_S [M + H]^+^ 883.3731; found 883.3779.

### 4.2. Plasmids

The cDNA encoding the rat mGlu_5b_R [25] was amplified by PCR using the following primers: FmGlu5NL (5′-AAACA**GAATTC**AGTGAGAGGAGGGTGGTGGCTC-3′) and RmGlu5NL (5′-AAAGA**TCTAGA**TCACAACGATGAAGAACTCTGCG-3′) and cloned into the EcoRI/XbaI sites of pNLF1-secN [CMV/Hygro] vector (Promega, Stockholm, Sweden), encoding a fusion of the secretory signal peptide sequence of IL-6 on the N terminus of NanoLuc (Nluc). The mGlu_5b_R^NL^ construct was confirmed by sequencing analysis. The resulting open reading frame therefore encoded a fusion of secreted Nluc at the N terminus of mGlu_5b_R. The mGlu_5b_R^SNAP^ construct was previously described [26].

### 4.3. Cell Culture and Transfection

Human embryonic kidney (HEK)-293T cells were grown in Dulbecco’s modified Eagle’s medium (DMEM) (Sigma-Aldrich, St. Louis, MO, USA) supplemented with 1 mM sodium pyruvate, 2 mM l-glutamine, 100 U/mL streptomycin, 100 mg/mL penicillin and 5% (*v*/*v*) fetal bovine serum at 37 °C and in an atmosphere of 5% CO_2_. HEK-293T cells growing in 60 cm^2^ plates were transfected with the cDNA encoding the different plasmids using linear PolyEthylenImine reagent (PEI) (Polysciences Inc., Valley Road Warrington, PA, USA) as previously described [27].

### 4.4. Membrane Preparations and Immunoblotting

To prepare membranes, cells were homogenized in ice-cold 50 mM Tris-HCl (pH 7.4), 1 mM EDTA, 300 mM KCl buffer containing a protease inhibitor cocktail (Roche Molecular Systems, Pleasanton, CA, USA) using a Polytron for three periods of 10 s each. The homogenate was centrifuged for 10 min at 1000× *g* at 4 °C. The resulting supernatant was centrifuged for 30 min at 12,000× *g* at 4 °C. Membranes were dispersed in 50 mM Tris HCl (pH 7.4) and 10 mM MgCl_2_ and protein concentration determined using the BCA protein assay kit (Thermo Fisher Scientific, Inc., Rockford, IL, USA). Sodium dodecyl sulfate-polyacrylamide gel electrophoresis (SDS/PAGE) was performed using 10% polyacrylamide gels. Proteins were transferred to Hybond^®^-LFP polyvinylidene difluoride (PVDF) membranes (GE Healthcare, Chicago, IL, USA) using a Trans-Blot^®^ SD Semi-Dry Transfer Cell (Bio-Rad, Hercules, CA, USA). PVDF membranes were blocked with 5% (wt/vol) dry non-fat milk in PBS containing 0.05% Tween-20 (PBS-T) during 2h and immunoblotted using rabbit anti-mGlu_5_R antibody (1 μg/mL; Millipore, Billerica, MA, USA) in blocking solution overnight at 4 °C. PVDF membranes were washed with PBS-T three times (5 min each) before incubation with a horseradish peroxidase (HRP)-conjugated goat anti-rabbit IgG (1/30,000; Pierce Biotechnology, Rockford, IL, USA) in blocking solution at 20 °C during 2 h. After washing the PVDF membranes with PBS-T three times (5 min each) the immunoreactive bands were developed using a chemiluminescent detection kit (Thermo Fisher Scientific) and detected with an Amersham Imager 600 (GE Healthcare).

### 4.5. NanoBRET Experiments

The NanoBRET assay was performed on cells transiently transfected with mGlu_5_R^NL^, according to [17]. In brief, cells were re-suspended in Hank’s Balanced Salt Solution (HBSS; 137 mM NaCl, 5.4 mM KCl, 0.25 mM Na2HPO4, 0.44 mM KH2PO4, 1.3 mM CaCl2, 1.0 mM MgSO4, 4.2 mM NaHCO3, pH 7.4), containing 10 mM glucose, and seeded into poli-ornitine coated white 96-well plates. After 24 h, cells were challenged with/without the non-labelled mGlu_5_R NAM (MTEP; Tocris Bioscience; Ellisville, MI, USA) and incubated for 1 h at 37 °C. Subsequently, the fluorescent ligand was added and the plate returned to 37 °C for 1 h. Finally, coelenterazine 400a (NanoLight Technologies; Pinetop, AZ, USA) was added at a final concentration of 1 μM, and readings were performed after 5 min using a CLARIOStar plate reader (BMG Labtech; Durham, NC, USA). The donor and acceptor emission was measured at 490 ± 10 nm and 650 ± 40 nm, respectively. The raw NanoBRET ratio was calculated by dividing the 650 nm emission by the 490 nm emission and the values fitted by non-linear regression using GraphPad Prism 7 (GraphPad Software, La Jolla, CA, USA). In competition studies, results were expressed as a percentage of the maximum signal obtained (mBU; miliBRET Units).

### 4.6. Intracellular Calcium Determinations

The mGlu_5_R-mediated intracellular Ca^2+^ accumulation was assessed by means of a luciferase reporter assay based in the expression of the nuclear factor of activated T-cells (NFAT), as previously described [28]. In brief, cells were transfected with the cDNA encoding the mGlu_5_R^SNAP^ and the NFAT-luciferase reporter (pGL4-NFAT-RE/luc2p; Promega). Plasmids were co-transfected by preparing a 1:1 solution (1.5 + 1.5 μg) in NaCl and mixing it with a solution containing 14.5 μg of PEI. The final solution was added to the cells plate for 4 h and thereafter replaced with fresh medium. After 36 h post-transfection, cells were first labelled with a non-permeable SNAP substrate (SNAP-Surface 647 ligand; New England BioLabs, Ipswich, MA, USA), as previously described [29]. Thereafter, cells were incubated with quisqualic acid in the presence/absence of the different NAMs tested for 6 h. Subsequently, cells were harvested with passive lysis buffer (Promega) and transferred to 96-wells white plates. First, fluorescence was assessed and thereafter the luciferase activity of cell extracts was determined using a luciferase Bright-GloTM assay (Promega) according to the manufacturer’s protocol. Both fluorescence (647 nm) and luciferase activity (535 nm) were determined in a CLARIOStar Optima plate-reader (BMG Labtech). The ratio between 535/647 nm was calculated.

### 4.7. Imaging

The localization of the mGlu_5_R NAMs was evaluated using a live-cell laser scanning confocal microscopy. To this end, cells were transfected with mGlu_5b_R^SNAP^ and, upon 36 h post-transfection, seeded into a chamber culture slide (Thermo Fisher Scientific). After 24 h, the medium was replaced by HBSS and single cells were examined using a Carl Zeiss LSM 880 spectral confocal laser scanning microscope (Carl Zeiss Microscopy GmbH, Jena, Germany) equipped with a multiline argon laser (458 nm, 488 nm and 514 nm), 405nm and 561 nm diode lasers and 633 nm He/Ne laser (Centres Científics i Tecnològics, Universitat de Barcelona, Bellvitge Campus, Barcelona, Spain) using a 63 × oil immersion objective (1.4 numerical aperture) an image resolution of 1024 × 1024 pixels. To assess association and diffusion of the fluorescent NAM (**4a**), it was added (5 μM) to the chamber and live-cell imaging was performed for 5 min.

### 4.8. Statistics

The number of samples (n) in each experimental condition is indicated in figure legends. Statistical analysis and significance is indicated for each experiment.

## 5. Conclusions

In summary, we have developed a new series of fluorescent ligands acting as NAMs for the mGlu_5_R. Interestingly, we showed that these ligands bind and modulate the receptor’s activity. In addition, as these they are compatible with NanoBRET binding experiments, they may eventually evolve into a useful tool for the pharmacological study of mGlu5R both in vitro and in vivo.

## Figures and Tables

**Figure 1 molecules-25-01532-f001:**
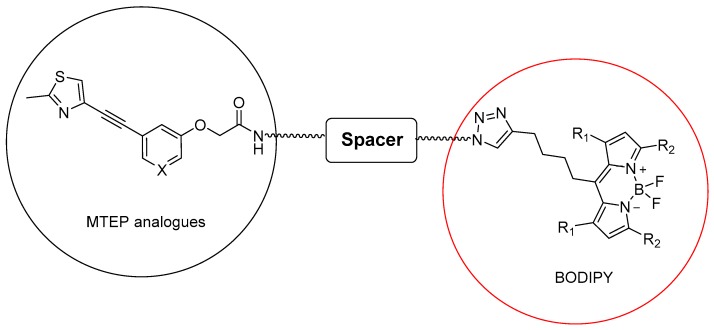
Schematic design of fluorescent mGlu_5_R ligands based on BODIPY.

**Figure 2 molecules-25-01532-f002:**
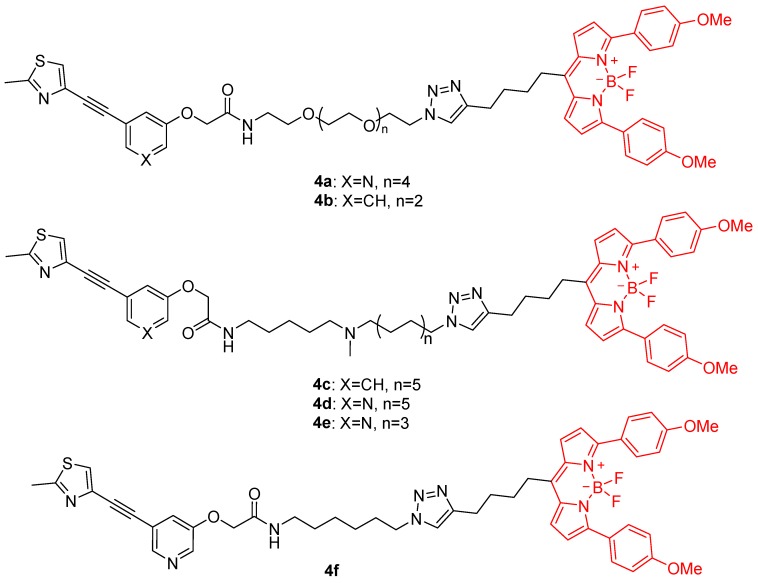
Structural design of the mGlu_5_R fluorescent ligands. Chemical representation of the six fluorescent NAMs (**4a**–**f**) in which a boron dipyrromethene (BODIPY; shown in red) fluorophore is fused with MTEP analogues by different spacers (shown in black).

**Figure 3 molecules-25-01532-f003:**
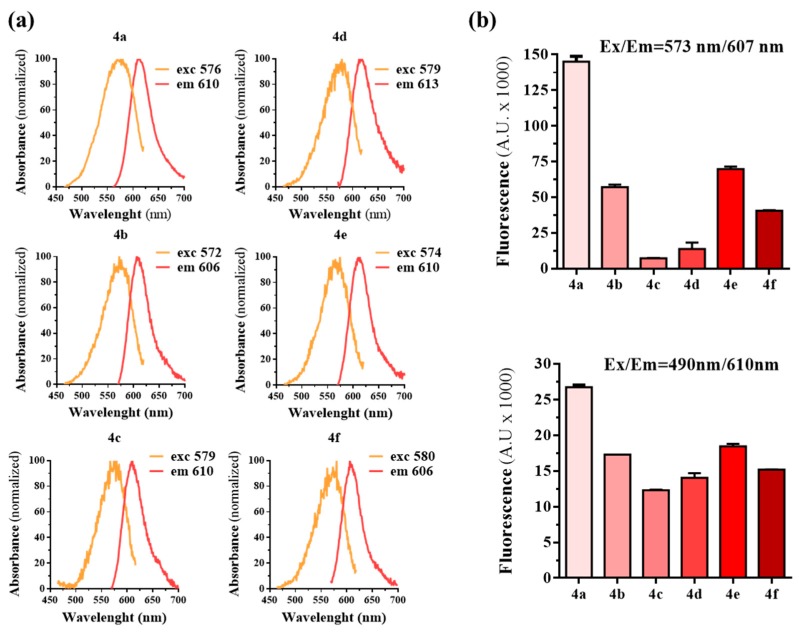
Characterization of the fluorescent properties of conjugates **4a**–**f**. (**a**) Excitation and emission spectra of the mGlu_5_R fluorescent ligands **4a**–**f**. The excitation/emission wavelength peak for each compound is shown in nanometers. (**b**) Fluorescence emission (arbitrary units, A.U.) at 607 nm or 610 nm is shown after exciting (573 nm or 490 nm, respectively) the six BODIPY-conjugates (10 µM solution in 0.05% DMSO of 4a to 4f depicted as light to dark red columns). This value is related to the quantum yield of each fluorophore.

**Figure 4 molecules-25-01532-f004:**
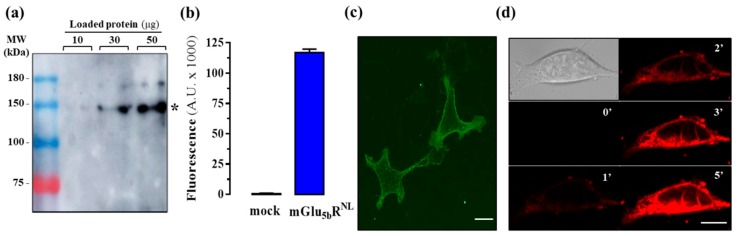
Evaluation of the mGlu_5b_R constructs and ligand **4a** in living cells. (**a**) Immunoblot detection of mGlu_5b_R^NL^. Increasing amounts of membrane extracts (10 μg, 30 μg and 50 μg of protein) from HEK293T cells transiently transfected with mGlu_5b_R^NL^ were analyzed by immunoblotting using a rabbit anti-mGlu_5_R antibody (1 μg/mL). Asterisk indicates the expected molecular weight (~150 kDa) of mGlu_5b_R^NL^. (**b**) Luminescence detection of mGlu_5b_R^NL^. HEK293T cells transiently transfected with mGlu_5b_R^NL^ were incubated with coelenterazine 400a for 5 min and luminescence recorded using a CLARIOStar plate reader. (**c**) The expression of mGlu_5b_R^SNAP^ was determined by assessing the fluorescence obtained after staining cells with the SNAP-Surface 647 substrate. (**d**) Ligand (**4a**) association to and diffusion in mGlu_5b_R^SNAP^ expressing cells as assessed by laser scanning confocal microscopy. The increase of the fluorescent signal was followed during 5 min. Scale bar: 100 μm.

**Figure 5 molecules-25-01532-f005:**
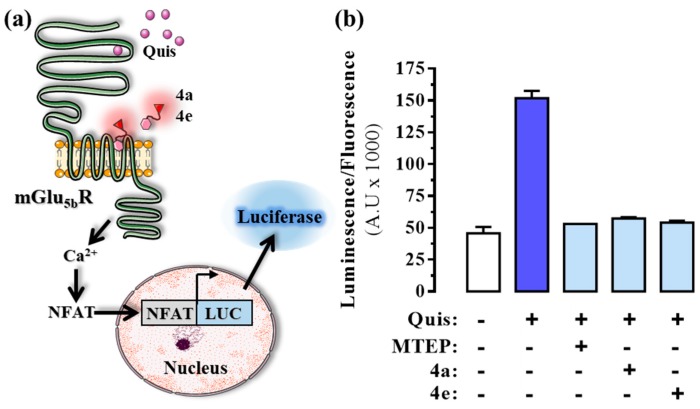
Effect of the fluorescent mGlu_5_R NAMs on intracellular calcium levels in HEK293T cells. (**a**) Schematic representation of the NFAT-luciferase reporter assay in which intracellular calcium levels were determined in the presence of quisqualic acid together with the fluorescent mGlu_5_R ligands **4a** and **4e**. (**b**) HEK293T cells transfected with mGlu_5b_R^SNAP^ were first labelled with a non-permeable SNAP substrate (SNAP-Surface 647). Next, cells were incubated with 100 µM of quisqualic acid (Quis) alone or in combination with 5 µM of MTEP, **4a** or **4e** for 6 h at 37 °C. The ratio between the luciferase activity (535 nm) and the fluorescence intensity of the receptor (647 nm) was calculated to normalize the signal by the number of receptor-expressing cells. Data are the mean ± SEM of three independent experiments performed in triplicate. ** *p* < 0.01, one-way ANOVA followed by Dunnett’s post-hoc test. Figure designed using image templates from [20].

**Figure 6 molecules-25-01532-f006:**
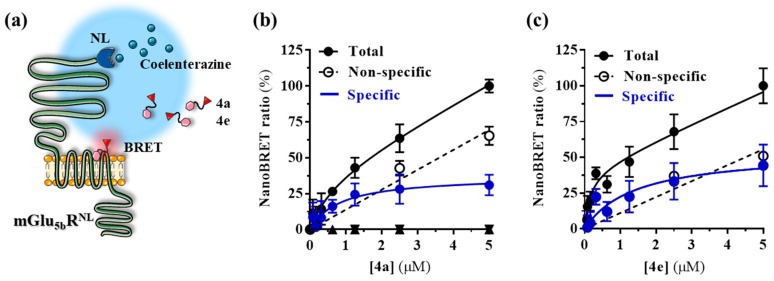
Determination of the binding affinity of **4a** and **4e** at mGlu_5b_R using the nanoBRET assay. (**a**) Illustrative representation of the nanoBRET assay in which a nanoluciferase linked to the N-terminal part of the mGlu_5b_R (donor) emits light at 490-10 nm in presence of coelenterazine. The light excites the BODIPY attached to the ligand (acceptor), which subsequently emits fluorescence at 650-80 nm. (**b**,**c**) NanoBRET saturation binding curves obtained by challenging mGlu_5b_R^NL^ expressing HEK293T cells with increasing concentrations of **4a** (**b**) or **4e** (**c**) in the absence (black circles) or presence (white circles) of 10 μM MTEP for 1 h at 37 °C. The specific binding curve (blue circles), the K_D_ and the B_max_ is shown for each ligand. The represented data are mean ± SEM of three independent experiments each performed in triplicate. Figure designed using image templates from [20].

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
