# Peer review of "Design, Synthesis and Characterization of a New Series of Fluorescent Metabotropic Glutamate Receptor Type 5 Negative Allosteric Modulators"

_molecules, 2020, doi:10.3390/molecules25071532_

Round 1

Reviewer 1 Report

The manuscript describes the synthesis and characterization of novel fluorescent allosteric modulators of the mGlu5R receptor and their application in nanoBRET assay. The topic is interesting and the manuscript demonstrates the feasibility of this approach in the development of allosteric modulators.

The first part of the results describe the chemical synthesis and characterization of the fluorescent properties of five BODIPY modified MTEP analogues. Due to the lack of competence in chemistry, I do not want to comment on the synthesis part of the manuscript.

In the next part the fluorescent characterization of the new compounds indicate a BODIPY peak excitation spectra at 572-580 nm and emission spectra maximums at 606-610 nm. Compound 4a and 4e were selected for further BRET experiments, because they showed the highest emission at 610 nm after excitation at 490 nm close to the peak emission wave length of NanoLuc. I have the following questions and observations considering this part of the manuscript.

1) There are lot of BODIPY compounds with different Ex/Em spectra. The compound used for conjugation in this experiments has peak excitation at 573 nm which is far away from the peak emission of NanoLuc. Why this BODIPY version was used instead of for example BODIPY 493/503,

which might be more suitable for the BRET experiments. The selection of this particular BODIPY version should be discussed.

2) In Figure 3b upper panel Ex/Em=570/630 nm is indicated. This is probably a misprint, both the text and figure legend say that in this experiment Ex/Em=573/607 was used.

3) In Figure 3 legend (line 121) „somehow related” does not sound right, should be rephrased.

The second part of the results describe the different cell based binding assays performed with the new compounds 4a and 4e. First the NanoLuc – mGlu5R construct is evaluated. Concerning this part, I have the following questions and observations.

4) In the cloning part of the methods section, it is described how the sequence coding the mGlu5R receptor was inserted into the NanoLuc vector, but information where it was cloned from is missing. Was the source an other vector or PCR amplified cDNA? If the source was PCR amplified DNA was it re-sequenced to check for mutations? Information whether it was the human receptor or from another species and which receptor isoform are also missing. For example, the homo sapiens glutamate metabotropic receptor 5 (GRM5), transcript variant b, mRNA (NCBI Reference Sequence: NM_000842.4) contains two NotI sites in the coding sequence, which makes it difficult to clone with NotI restriction endonuclease. A much more detailed description on what was cloned into the NanoLuc vector is required in this section.

5) In Figure 4a upper panel immunoblotting of membranes of mGlu5R expressing cells is presented. For me it is not completely clear if this photo represents a Western-blot or a dot blot. It looks like a dot blot with the rounded signals, but the method section describes Western-blot and the 150 kDa marker also suggest Western-blot. Also six signals can be observed, but only three concentrations are indicated. If the samples were duplicated it should be indicated in the figure legend. Due to this discrepancies I suggest that a bigger part of the blot is included into the figure for clarification, where possibly the presence of other bands can be evaluated (for example at 130 KDa at the size of the native receptor).

6) In line 129 „no signal was not”, should be changed to „no signal was observed”.

7) In line 131 „sine qua no” should be changed to „sine qua non”.

In the next section, experiments with cells expressing the mGlu5R-SNAP fusion protein is described.

However, no information at all is given about the used expression vector, exactly which receptor it contained. This information is important, because next the modulation of receptor activation is tested with this construct.

8) A detailed description of the mGlu5R-SNAP vector and the transient transfection protocol should be included in the methods section.

Next negative allosteric modulation of the new compounds were tested in complex assay based on NFAT activation of luciferase. Both new compounds showed potent negative modulation of the receptor similarly to MTEP. My only concern with the experiment is that it is necessary to use a fluorescent correction step due to the transient nature of the transfection. It would have been better to use a stably transfected cell line with uniform receptor expression. Also the fluorescent SNAP labeling only indicates the presence of the receptor, not the luciferase reporter construct. Obviously, the construction of such a cell line takes long time, so the use of a transient system is understandable.

9) Nevertheless, the parameters (method used, amount of vectors) of the cotransfection with the two vectors should be provided in the method section.

10) In Figure 5b the labeling of the Y axis should be either luminescence or luminescence/fluorescence

certainly not fluorescence, because in this experiment the level of luciferase expression indicated by the luminescent signal is presented.

In the final experiments the 4a and 4e compounds are tested in the nanoBRET assay and their binding parameters are determined. Just as described previously, a stable cell line expressing the mGlu5R-NanoLuc construct would be more suitable for these experiments. Other observations and questions regarding these experiments:

11) In line 165 the binding curves are described as „bell-shaped” which is incorrect, the presented curves are standard saturation binding rectangular hyperbolas. This should be corrected.

12) In these experiments coelenterazin 400a was used, which is a blue shifted version with emission max at 411 nm very far from the maximum excitation of the ~570 nm of compound 4a and 4e. Why the natural coelenterazin with Exmax=509 was not used?

13) In the legend of Figure 5 and the method section it is indicated that the fluorescence emission is measured at 650-80 nm. It is not clear for me what does this notation mean? Does this band cover the 610 nm emission maximum of the compounds like 650±80 nm or it is more like 650±40 nm? If it is the later why the emission was not measured closer to the maximum of the new compounds?

14) In the legend of Figure 5 it is indicated that Kd and Bmax values are indicated for the ligands. Bmax values are not indicated in either the graph or the text, which should be corrected.

15) Error bars are missing from the Total curve in graph two, they should be added.

16) In line 188 „As their remains” should be „As remains”.

17) In line 190 „clear” should be changed to „clearly”.

Author Response

The manuscript describes the synthesis and characterization of novel fluorescent allosteric modulators of the mGlu5R receptor and their application in nanoBRET assay. The topic is interesting, and the manuscript demonstrates the feasibility of this approach in the development of allosteric modulators.

We thank the Reviewer for the positive and constructive comments to improve the manuscript.

The first part of the results describes the chemical synthesis and characterization of the fluorescent properties of five BODIPY modified MTEP analogues. Due to the lack of competence in chemistry, I do not want to comment on the synthesis part of the manuscript.

In the next part the fluorescent characterization of the new compounds indicates a BODIPY peak excitation spectra at 572-580 nm and emission spectra maximums at 606-610 nm. Compound 4a and 4e were selected for further BRET experiments, because they showed the highest emission at 610 nm after excitation at 490 nm close to the peak emission wavelength of NanoLuc. I have the following questions and observations considering this part of the manuscript.

1) There are lot of BODIPY compounds with different Ex/Em spectra. The compound used for conjugation in this experiment has peak excitation at 573 nm which is far away from the peak emission of NanoLuc. Why this BODIPY version was used instead of for example BODIPY 493/503, which might be more suitable for the BRET experiments. The selection of this particular BODIPY version should be discussed.

The Reviewer raises an interesting point, which may be relevant to include in the discussion section. Thus, theoretically, BODIPY dyes with lower excitation peaks would be desirable to be excited by Nanoluc emission. Indeed, we already synthesized a series of BODIPY-conjugates, with excitation at 493 nm and emission at 503 nm. However, when used in BRET experiments, we failed to obtain a specific BRET signal. The reason was that NanoLuc emission is very high and it overlaps with the BRET signal, masking it. Thus, it is preferable to detect BRET signals at higher wavlenghts (>610 nm), in which Nanoluc emission is very much lower. We have included the following statement in the text: “it is important to note that we used BODIPY derivatives with excitation/emission peaks at 573/607 nm. The reason is that NanoLuc exhibits a high emission upon coelenterazine incubation, and this signal overlaps and mask specific BRET signals when using dyes with lower excitation/emission wavelengths (data not sho wn)”.

2) In Figure 3b upper panel Ex/Em=570/630 nm is indicated. This is probably a misprint; both the text and figure legend say that in this experiment Ex/Em=573/607 was used.

The Reviewer is right. We have corrected the misprint.

3) In Figure 3 legend (line 121) „somehow related” does not sound right, should be rephrased.

We have suppressed the “somehow”, we agree it does not sound right.

The second part of the results describe the different cell based binding assays performed with the new compounds 4a and 4e. First the NanoLuc – mGlu5R construct is evaluated. Concerning this part, I have the following questions and observations.

4) In the cloning part of the methods section, it is described how the sequence coding the mGlu5R receptor was inserted into the NanoLuc vector, but information where it was cloned from is missing. Was the source another vector or PCR amplified cDNA? If the source was PCR amplified DNA was it re-sequenced to check for mutations? Information whether it was the human receptor or from another species and which receptor isoform are also missing. For example, the homo sapiens glutamate metabotropic receptor 5 (GRM5), transcript variant b, mRNA (NCBI Reference Sequence: NM_000842.4) contains two NotI sites in the coding sequence, which makes it difficult to clone with NotI restriction endonuclease. A much more detailed description on what was cloned into the NanoLuc vector is required in this section.

We apologize for the curtailed information provided while describing the mGlu5R-NL construct generation. In addition, thanks to the reviewer’s observation we noticed that the description of the cloning was inadequate. Thus, the NotI restriction enzyme site was not used in this specific cloning, it was used for the generation of another related mGlu5R construct performed for the very same person whom confound the cloning description. We are so sorry for this mistake. Therefore, we amended accordingly: “The cDNA encoding the rat mGlu5bR [23] was amplified by PCR using the following primers: FmGlu5NL (5′-AAACAGAATTCAGTGAGAGGAGGGTGGTGGCTC-3′) and RmGlu5NL (5′-AAAGATCTAGATCACAACGATGAAGAACTCTGCG-3′) and cloned into the EcoRI/XbaI sites of pNLF1-secN [CMV/Hygro] vector (Promega, Stockholm, Sweden), encoding a fusion of the secretory signal peptide sequence of IL-6 on the N terminus of NanoLuc (Nluc). The mGlu5bRNL construct was confirmed by sequencing analysis. The resulting open reading frame therefore encoded a fusion of secreted Nluc at the N terminus of mGlu5bR. The mGlu5bRSNAP construct was previously described [24].”

5) In Figure 4a upper panel immunoblotting of membranes of mGlu5R expressing cells is presented. For me it is not completely clear if this photo represents a Western-blot or a dot blot. It looks like a dot blot with the rounded signals, but the method section describes Western-blot and the 150 kDa marker also suggest Western-blot. Also, six signals can be observed, but only three concentrations are indicated. If the samples were duplicated it should be indicated in the figure legend. Due to this discrepancy I suggest that a bigger part of the blot is included into the figure for clarification, where possibly the presence of other bands can be evaluated (for example at 130 KDa at the size of the native receptor).

We apologize for the low quality of the immunoblot shown in Figure 4. As suggested by the Reviewer we include now a bigger image for clarification. Thus, although it seems clear that the sample ran in a non-very homogenous way, it can be clearly seen that the construct was expressed at the predicted molecular weight.

6) In line 129 „no signal was not”, should be changed to „no signal was observed”.

Following the indications of the Reviewer we have changed the sentence.

7) In line 131 „sine qua no” should be changed to „sine qua non”.

Following the indications of the Reviewer we have corrected the mistake.

In the next section, experiments with cells expressing the mGlu5R-SNAP fusion protein is described.

However, no information at all is given about the used expression vector, exactly which receptor it contained. This information is important, because next the modulation of receptor activation is tested with this construct.

8) A detailed description of the mGlu5R-SNAP vector and the transient transfection protocol should be included in the methods section.

We apologize for the inconvenience. Thus, in the new version of the manuscript we include the reference describing the generation of the mGlu5bR-SNAP (i.e. Morató, X.; Luján, R.; Gonçalves, N.; Watanabe, M.; Altafaj, X.; Carvalho, A. L.; Fernández-Dueñas, V.; Cunha, R. A.; Ciruela, F. Metabotropic glutamate type 5 receptor requires contactin-associated protein 1 to control memory formation. Human Molecular Genetics 2018, 27, 3528–3541). In addition, we provide the reference describing the PEI transfection method (i.e. Longo, P. A.; Kavran, J. M.; Kim, M.-S.; Leahy, D. J. Transient mammalian cell transfection with polyethylenimine (PEI). Methods in Enzymology 2013, 529, 227–240).

Next negative allosteric modulation of the new compounds was tested in complex assay based on NFAT activation of luciferase. Both new compounds showed potent negative modulation of the receptor similarly to MTEP. My only concern with the experiment is that it is necessary to use a fluorescent correction step due to the transient nature of the transfection. It would have been better to use a stably transfected cell line with uniform receptor expression. Also the fluorescent SNAP labeling only indicates the presence of the receptor, not the luciferase reporter construct. Obviously, the construction of such a cell line takes long time, so the use of a transient system is understandable.

We agree with the Reviewer that stable cell lines are often desirable compared with transient expression. On the other hand, these transients allowed us to use different constructs to conduct both binding and functional experiments. Indeed, the SNAP construct was especially valuable to normalize the luminescence obtained, since only membrane receptors are labelled when using a non-permeable dye.

9) Nevertheless, the parameters (method used, amount of vectors) of the cotransfection with the two vectors should be provided in the method section.

Following the indications of the Reviewer we have provided details regarding the transfection parameters: “Plasmids were co-transfected by preparing a 1:1 solution (1.5+1.5 μg) in NaCl and mixing it with a solution containing 14.5 μg of PEI. The final solution was added to the cells plate for 4 h and thereafter replaced with fresh medium”.

10) In Figure 5b the labeling of the Y axis should be either luminescence or luminescence/fluorescence certainly not fluorescence, because in this experiment the level of luciferase expression indicated by the luminescent signal is presented.

The Reviewer is right. We have corrected the mistake.

In the final experiments the 4a and 4e compounds are tested in the nanoBRET assay and their binding parameters are determined. Just as described previously, a stable cell line expressing the mGlu5R-NanoLuc construct would be more suitable for these experiments. Other observations and questions regarding these experiments:

11) In line 165 the binding curves are described as „bell-shaped” which is incorrect, the presented curves are standard saturation binding rectangular hyperbolas. This should be corrected.

Following the indications of the Reviewer we have suppressed “bell-shaped”.

12) In these experiments coelenterazin 400a was used, which is a blue shifted version with emission max at 411 nm very far from the maximum excitation of the ~570 nm of compound 4a and 4e. Why the natural coelenterazin with Exmax=509 was not used?

In our experiments, upon incubation with coelenterazine 400a, NanoLuc exhibits a peak around 465 nm. It is true that the intensity at 570 nm is much lower than at 465 nm (about 8 times) but it is enough to excite the BODIPY dye and, as explained above, it does not mask the specific BRET signal obtained at wavelengths higher than 610 nm. A similar approach was previously reported (Stoddard et al., Nature methods, 2015,12:661-663). On the other hand, it is important to note that coelenterazine 400a is a substrate that allows a stable emission signal of NanoLuc for longer periods than the natural one, thus minimizing variability between different assays (Kobayashi et al., Nature Protocols, 2019,14:1084-1107).  

13) In the legend of Figure 5 and the method section it is indicated that the fluorescence emission is measured at 650-80 nm. It is not clear for me what does this notation mean? Does this band cover the 610 nm emission maximum of the compounds like 650±80 nm or it is more like 650±40 nm? If it is the later why the emission was not measured closer to the maximum of the new compounds?

We have clarified the emission wavelengths in the text: 650±40 nm. As previously commented, we selected these wavelengths based on previous reports (Stoddard et al., Nature methods, 2015,12:661-663), since this approach allows to minimize the overlapping of NanoLuc emission into the specific BRET signal.

14) In the legend of Figure 5 it is indicated that Kd and Bmax values are indicated for the ligands. Bmax values are not indicated in either the graph or the text, which should be corrected.

The Reviewer is right. We have included Bmax values in the text.

15) Error bars are missing from the Total curve in graph two, they should be added.

We have added the error bars. We are sorry for the mistake.

16) In line 188 „As their remains” should be „As remains”.

Following the indications of the Reviewer we have corrected the mistake.

17) In line 190 „clear” should be changed to „clearly”.

Following the indications of the Reviewer we have corrected the mistake.

Reviewer 2 Report

The submitted manuscript entitled “Design, synthesis and characterization of a new series of fluorescent metabotropic glutamate receptor type 5 negative allosteric modulators” presents a new group of fluorescent ligands that can be used in the new NanoBRET-based binding assay, which will facilitate screening for new mGlu5R allosteric modulators.

The presented manuscript is interesting and has a practical dimension. Authors developed a new series of fluorescent ligands acting as NAMs for the mGlu5R, and showed that they bind and modulate the receptor’ activity. As a result, they can be a useful tool for pharmacological studies of mGlu5R both in vitro and in vivo. As the authors point out, the developed nanoBRET binding assay allowed reliable assessment of mGlu5R allosteric ligand binding, and can be seen as a useful tool for developing new mGlu5R allosteric ligands.

In my opinion, presented manuscript can be accepted for publication after a few corrections.

In Reference section:

There is no detailed information about the volume numbers and pages for many cited works, including 2, 3, etc.

Ref. [25] there is no citation in the paper.

Author Response

The submitted manuscript entitled “Design, synthesis and characterization of a new series of fluorescent metabotropic glutamate receptor type 5 negative allosteric modulators” presents a new group of fluorescent ligands that can be used in the new NanoBRET-based binding assay, which will facilitate screening for new mGlu5R allosteric modulators.

The presented manuscript is interesting and has a practical dimension. Authors developed a new series of fluorescent ligands acting as NAMs for the mGlu5R and showed that they bind and modulate the receptor’ activity. As a result, they can be a useful tool for pharmacological studies of mGlu5R both in vitro and in vivo. As the authors point out, the developed nanoBRET binding assay allowed reliable assessment of mGlu5R allosteric ligand binding and can be seen as a useful tool for developing new mGlu5R allosteric ligands.

In my opinion, presented manuscript can be accepted for publication after a few corrections.

We thank the Reviewer for the positive comments.

In Reference section:

There is no detailed information about the volume numbers and pages for many cited works, including 2, 3, etc.

We are sorry about that, but while formatting the references with Mendeley we have lost some information. Thus, we amended accordingly.

Ref. [25] there is no citation in the paper.

The Reviewer was right. We have revised the reference list.

Reviewer 3 Report

The study by Fernández-Dueñas, Qian et al., describes the elegant synthesis of a series of fluorescently labeled allosteric modulators of mGluR5 and characterized their binding efficiency in cells using a nanoBRET assay. Overall, the experiments are well designed and described in sufficient detail. This set of novel ligands and the described assays are very innovative and will be useful for further development and screening of novel mGluR5 allosteric modulators.

I have a few minor comments that are meant to improve the manuscript before acceptance:

  • The labeling experiments in Figure 4b and c suggest that the ligands label the plasma membrane and over time accumulate in the cell. First, I am not sure why this is “in concordance with lipophilic nature”? Would this not simply be expected from binding to a receptor on the cell membrane? Second, the images indeed suggest that the ligands are taken up by the cell. The authors suggest this is by adsorption, but could it not also be via endocytosis of the bound receptor?
  • The protein structures used in figure 5a and 6a are illustrating the nanoBRET principle, but are a bit misleading. mGluR5 is in fact much larger, only the transmembrane domains are presented, and it seems that for NanoLuc the structure of GFP is taken. So, although I appreciate the effort to illustrate the nanoBRET principle, since the scale (and structure) is off tremendously, it might be misleading as the authors rightly point out that NanoLuc is almost 10x smaller than mGluR5. Either a cartoon depiction, roughly at scale, or the correct structures would improve the illustration.
  • In figure 3b, the reported wavelengths in the above the graphs do not correspond with the reported wavelengths in the legend. Upper panel: “Ex/Em =570 nm/630nm”, lower panel: “Ex/Em = 490nm/610nm”, while in the figure legend it says “… at 607 nm or 610 nm is shown after exciting (573 nm or 490 nm, respectively)”
  • Line 131: “sine qua no” should be “sine qua non”
  • Figure 4b and c miss a scale bar
  • Line 190: “clear” should be “clearly”?
  • Line 199: “binging-assay” should be “binding-assay”?
  • Line 209: “penetrated at the plasma membrane…” should be “penetrated the plasma membrane”

Author Response

The study by Fernández-Dueñas, Qian et al., describes the elegant synthesis of a series of fluorescently labeled allosteric modulators of mGluR5 and characterized their binding efficiency in cells using a nanoBRET assay. Overall, the experiments are well designed and described in sufficient detail. This set of novel ligands and the described assays are very innovative and will be useful for further development and screening of novel mGluR5 allosteric modulators.

We thank the Reviewer for the positive comments.

I have a few minor comments that are meant to improve the manuscript before acceptance:

The labeling experiments in Figure 4b and c suggest that the ligands label the plasma membrane and over time accumulate in the cell. First, I am not sure why this is “in concordance with lipophilic nature”? Would this not simply be expected from binding to a receptor on the cell membrane? Second, the images indeed suggest that the ligands are taken up by the cell. The authors suggest this is by adsorption, but could it not also be via endocytosis of the bound receptor?

The Reviewer raises an interesting point here. On the one hand, it seems clear that together with the binding to the receptor, some adsorption would happen, as described previously (Cottet et al., Biochemical Society transactions 2013, 41, 148–153). And, thereafter, the ligand seems to penetrate the cell. As the Reviewer claims, we don’t know the mechanism by which the ligand reaches the cytoplasm, and we should also consider endocytosis. Accordingly, we have revised the text: “The fact that the ligand penetrated into the cell, either by diffusion or endocytosis, supported the need…” 

The protein structures used in figure 5a and 6a are illustrating the nanoBRET principle, but are a bit misleading. mGluR5 is in fact much larger, only the transmembrane domains are presented, and it seems that for NanoLuc the structure of GFP is taken. So, although I appreciate the effort to illustrate the nanoBRET principle, since the scale (and structure) is off tremendously, it might be misleading as the authors rightly point out that NanoLuc is almost 10x smaller than mGluR5. Either a cartoon depiction, roughly at scale, or the correct structures would improve the illustration.

Following the indications of the Reviewer we have improved the illustrations of Fig 5a and 6a. Now, cartoons with a roughly scaled mGlu5R structure are included.

In figure 3b, the reported wavelengths in the above the graphs do not correspond with the reported wavelengths in the legend. Upper panel: “Ex/Em =570 nm/630nm”, lower panel: “Ex/Em = 490nm/610nm”, while in the figure legend it says “… at 607 nm or 610 nm is shown after exciting (573 nm or 490 nm, respectively)”

We apologize for the misprint, thus we amended accordingly.

Line 131: “sine qua no” should be “sine qua non”

Following the indications of the Reviewer we have corrected the mistake.

Figure 4b and c miss a scale bar

We thank the reviewer for the observation, thus we include the scale bar.

Line 190: “clear” should be “clearly”?

Following the indications of the Reviewer we have corrected the mistake.

Line 199: “binging-assay” should be “binding-assay”?

Following the indications of the Reviewer we have corrected the mistake.

Line 209: “penetrated at the plasma membrane…” should be “penetrated the plasma membrane”

Following the indications of the Reviewer we have corrected the mistake.

Round 2

Reviewer 1 Report

Dear Authors!

Thank you for the clarifications and corrections. I am satisfied with all the explanations and recommend the manuscript for publication.

Good luck with your next work and keep safe